# Recent Developments in Two-Dimensional Carbon-Based Nanomaterials for Electrochemical Water Oxidation: A Mini Review

Yuxin Zhao [1,3,†], Siyuan Niu [1,3,†], Baichuan Xi [2], Zurong Du [1,3], Ting Yu [1,3], Tongtao Wan [1,3,*], Chaojun Lei [2,*] and Siliu Lyu [1,3,*]

1 Hubei Key Laboratory of Automotive Power Train and Electronic Control, Shiyan 442002, China; 202204070@huat.edu.cn (Y.Z.); 202204071@huat.edu.cn (S.N.); 20230125@huat.edu.cn (Z.D.); yuting@huat.edu.cn (T.Y.)
2 College of Material, Chemistry and Chemical Engineering, Key Laboratory of Organosilicon Chemistry and Material Technology, Ministry of Education, Hangzhou Normal University, Hangzhou 311121, China; 2022112009042@stu.hznu.edu.cn
3 School of Automotive Engineering, Hubei University of Automotive Technology, Shiyan 442002, China
* Correspondence: 20230056@huat.edu.cn (T.W.); chaojun_lei@hznu.edu.cn (C.L.); lsl1102@huat.edu.cn (S.L.)
† These authors contributed equally to this work.

**Abstract:** Water splitting is considered a renewable and eco−friendly technique for future clean energy requirements to realize green hydrogen production, which is, to a large extent, hindered by the oxygen evolution reaction (OER) process. In recent years, two−dimensional (2D) carbon−based electrocatalysts have drawn sustained attention owing to their good electrical conductivity, unique physicochemical properties, and excellent electrocatalytic performance. Particularly, it is easy for 2D carbon−based materials to form nanocomposites, which further provides an effective strategy for electrocatalytic applications. In this review, we discuss recent advances in synthetic methods, structure−property relationships, and a basic understanding of electrocatalytic mechanisms of 2D carbon−based electrocatalysts for water oxidation. In detail, precious, non−precious metal−doped, and non−metallic 2D carbon−based electrocatalysts, as well as 2D carbon−based confined electro-catalysts, are introduced to conduct OER. Finally, current challenges, opportunities, and perspectives for further research directions of 2D carbon−based nanomaterials are outlined. This review can provide significant comprehension of high−performance 2D carbon−based electrocatalysts for water-splitting applications.

**Keywords:** 2D carbon−based electrocatalysts; oxygen evolution reaction; overpotential; active sites; electrocatalytic mechanism





## 1. Introduction

Due to the impending depletion of fossil fuels and growing levels of environmental pollution, developing sustainable and clean energy has become an important exploration direction [1]. Under the circumstances, appropriate substitutes to alleviate the reliance on fossil fuels, such as nuclear energy, wind energy, and hydrogen energy, have been developed. Hydrogen energy possesses many unique characteristics, such as being environmentally friendly and having abundant reserves and high energy density, and thus is regarded as a promising candidate for the development of low-carbon economies [2]. Global hydrogen demand has increased annually from 59 ~Mt in 2000 to 88 ~Mt in 2020, and this demand is forecast to increase to 211 ~Mt in 2030 and to 528 ~Mt in 2050. In developing the hydrogen value chain, a certain amount of investment is necessary, but at the same time, the hydrogen economy will also bring income [3].

In recent years, a variety of production technologies have been developed to crack water molecules and release hydrogen. Among them, hybrid water electrolysis (HWE), com-

bining the thermodynamically favorable OER processes at the anode with the cathodic HER processes, is an attractive solution for increasing the yield of $H_2$ [4]. Compared with traditional approaches, heterointerface engineering, a potential way to design high-performance nanomaterials, has the characteristics of adjustable electronic structure, regulated dynamics, enhanced stability, and electrochemical activity, which gives heterointerface engineering the advantages of rich implementation means, broad action scope, and superior electrochemical effect [5]. However, electrocatalytic water splitting is cost-effective, convenient, and environmentally friendly and, therefore, has great potential to produce hydrogen at the cathode and oxygen at the anode [6,7]. Generally speaking, electrocatalytic water splitting can be divided into two half−reactions, which are the hydrogen evolution reaction (HER) and oxygen evolution reaction (OER), respectively [8]. Compared with HER, OER has slow kinetics involving the transfer process with four electrons and four protons, which needs higher thermodynamic potential to overcome [9,10]. The larger voltage that the transfer process requires hinders the overall efficiency of the water−splitting reaction. Two-dimensional (2D) materials have been widely studied for electrocatalysts in the field of renewable energy. A wide variety of 2D materials for energy conversion and storage systems has been discovered [11,12]. The 2D materials have better bending flexibility and atomic thickness combined with higher in−plane strength and stiffness as compared to traditional 1D and 3D materials [13,14]. Due to their special planar structure with atomic thickness, 2D materials have obvious benefits to catalyze water oxidation, such as possessing a larger specific surface area and giving a wealth of exposed active sites, making them easy to combine, and showing excellent catalytic activity through the introduction of defects or heteroatoms [15]. Among 2D materials, 2D MXene materials have good metallic conductivity and are hydrophilic, which makes them ideal for electrocatalysis. However, pristine MXenes are difficult to use as an electrocatalyst directly, because of their low catalytic activity. The 2D MOFs have an ultra−thin thickness, different arrangements of surface atomic bonding, and a high degree of exposed catalytic active sites. Nonetheless, due to their inherent molecular structure, most MOF materials have poor electrical conductivity compared with other materials. It is remarkable that 2D carbon−based materials have become a new star in the field of electrocatalytic water decomposition due to their advantages, such as their low cost, adjustable molecular structure, and strong acid/alkaline resistance [15]. The 2D carbon−based electrocatalysts are composed of single or multiple atoms doped with carbon material by various methods. In 2018, Zhang et al. used the template method to prepare an electrocatalyst (VCNs@FeOOH) formed with vertically aligned carbon nanosheets (VCNs) and iron oxyhydroxide/nitride (FeOOH/FeN$_4$), and the FeOOH/FeN$_4$ was verified to have high activity and excellent durability [16]. In 2017, Lei et al. adopted the CPT method to regulate the surface functional group composition of carbon materials by applying cathodic polarization treatment (CPT) of different durations, and then, the carbon material was dried overnight in a vacuum oven to obtain ZIF−8−C0 [17]. In 2019, Zhang et al. synthesized a new class of Co@N−C materials (C−MOF−C2−T) by using a MOF−derived method with raw MOF material, and the C−MOF−C2-900 was found to have good electrocatalytic properties [18]. At the same time, other effective methods have also been applied to the synthesis of two−dimensional carbon-based water splitting catalysts, such as carbonization [19,20], chemical vapor deposition (CVD) [21,22], hydrothermal [23,24], solvothermal [25,26], the pyrolysis method [27,28], and so on. This review mainly summarizes the materials obtained by doping the atoms of precious metals (Ir, Ru, Rh), non−precious metals (Fe, Ni, Co), and non−metals (N, S, P, F) with carbon materials (graphyne, graphene, carbon nanosheets, carbon cloth, etc.). The main doping methods include pyrolysis, solvothermal, the salt mode method, hydrothermal, the in-situ reduction method, etc. The 2D carbon−based materials have a nanostructured conductive network that facilitates electron transport and abundant pores to provide a large surface area and enhance mass transport facilities as well as expose more active sites for OER progress. Meanwhile, as a very important catalyst support for metals and metal−derived materials, 2D carbon−based materials usually show improving efficiency and provide

more reachable active sites [29,30]. Most of the OER progresses were carried out under alkaline conditions and can also be carried out under given acidic conditions. The OER electrocatalysts can be applied to alkaline electrolyzers and metal−air batteries in alkaline conditions, and the application of the technology in hydrogen fuel production is relatively mature [31].

Until now, some 2D carbon−based materials−related reviews have been reported with emphasis on synthesis, structure, and potential applications. Notwithstanding, less attention has been paid to the recent developments of 2D carbon−based materials for OER in terms of the synthetic method, performances, and the reasons for high activity. In this review, we summarize recent research work in regard to the synthetic methods, OER performances, and the leading factors of 2D carbon−based high−performing electrocatalysts. Since many reviews have been centralized on the OER electrocatalysts, including other dimensional structures, this review mainly focuses on the discussion about the OER performances of 2D carbon−based electrocatalysts in recent years using performance comparison, synthetic method, surface area, and stability. Four classes of 2D carbon−based catalysts are introduced, including precious metal−doped 2D carbon−based electrocatalysts, non−precious metal−doped 2D carbon−based electrocatalysts, non−metallic 2D carbon−based electrocatalysts, and 2D carbon−based confined electrocatalysts. In the final section, the major challenges and perspectives, including the development of applications and the reaction descriptors for further study on 2D carbon−based electrocatalysts, are outlined. This review will offer a short but significant reference for the researchers to address the recent advances in the challenges and rationally design high−performing 2D carbon-based nanomaterials to conduct OER.

## 2. Precious Metal−Doped 2D Carbon-Based Electrocatalysts

Currently, Ir−based and Ru−based materials are known as the most advanced and efficient OER electrocatalysts [32,33]. Nevertheless, their restricted multifunctional performance and high cost limit their large−scale applications for OER [34,35]. Therefore, it is necessary to develop efficient and stable OER electrocatalysts to promote the development of relevant renewable energy equipment [36–38]. In the precious metal group, ruthenium (Ru) has excellent catalytic activity and a lower cost; therefore, Ru−based nanocomposites and Ru−based nanocomposites compounded with a second metal have been widely used for efficient electrocatalysis in water−splitting systems [39–41]. The structure of precious metal−doped 2D carbon−based electrocatalysts is shown in Figure 1a. Next, we will introduce several relevant research achievements, such as Ru@g−CN$_x$, Ru-G/CC−350, Co−Ru@RuO$_x$/NCN, CoRuO$_x$@C, Rh−GO, and Ir−IrO$_x$/C−20.

Gao et al. injected RuCl$_3$·H$_2$O and other necessary materials into a Radleys Carousel reactor tube. Then, the mixture was degassed in N$_2$ and, in an inert atmosphere, 180 °C refluxed 72 h to obtain a precipitate. Then, the precipitate was separated, washed, and dried at room temperature to synthesize power Ru@CIN−1. In the same way, CIN−1 was prepared without RuCl$_3$·H$_2$O and 2−(diphenylphosphino) benzaldehyde. Finally, under an N$_2$ atmosphere at 500 °C for 2 h, Ru@CIN−1 and pure CIN−1 were calcined to obtain a new functional ruthenium catalyst g−CN$_x$ and Ru@g−CN$_x$ with a layered−sheet structure, respectively. The LSV curves of Ru@g−CN$_x$, RuO$_2$, Ru@CIN−1, and g−CN$_x$ show that at a benchmark current density of 10 mA cm$^{-2}$, the Ru@gCN$_x$ gives an overpotential of 280 mV. The Tafel slope values of commercial RuO$_2$ (74.3 mV dec$^{-1}$), Ru@CIN−1 (381.8 mV dec$^{-1}$), and g−CN (235.6 mV dec$^{-1}$) are higher than that of Ru@g−CN$_x$, which is around 49.5 mV dec$^{-1}$. Moreover, the RuO$_2$/N−C composites show excellent overall water−splitting performance that surpass the commercial Pt/C and RuO$_2$ couple. The superior performance could be attributed to the small size of RuO$_2$ and the synergy of N−C and RuO$_2$ [42]. In 2022, Chen's group synthesized a novel Ru−based electrocatalyst with Ru/RuO$_2$ heterostructure via a glycerol−assisted solvothermal strategy (Figure 2a,b) [43]. To avoid the reduction in the Ru utilization rate induced by agglomeration, carbon cloth as a conductive substrate was introduced to enhance the cohesion

and electrical conductivity between the catalysts and the substrates. The electrocatalytic OER activities of Ru−G/CC−350 and annealed sample of Ru−H$_2$O/CC−350 in 1 M KOH were investigated [44]. Compared to the Ru−G/CC−350 and RuO$_2$, the as-prepared Ru−H$_2$O/CC−350 displays a current density of 10 mA cm$^{-2}$ with a lowest overpotential of 270 mV (Figure 2c). It can be seen from Figure 2d that the Tafel slope of Ru−H$_2$O/CC−350 shows the lowest value of 63 mV dec$^{-1}$, which further confirms that Ru−H$_2$O/CC−350 has the highest electrocatalytic activity. The chronoamperometry of Ru−H$_2$O/CC−350 is evaluated at the overpotential of 270 mV for 100 h (Figure 2e) to further affirm the OER stability of Ru−H$_2$O/CC−350. An Ru/RuO$_2$ heterostructure that can determine the rate of OER process was formed by oxidizing amorphous Ru. The excellent performance of Ru−H$_2$O/CC−350 could be ascribed to the formed Ru/RuO$_2$ heterostructure with ample defects of oxygen vacancies [43].

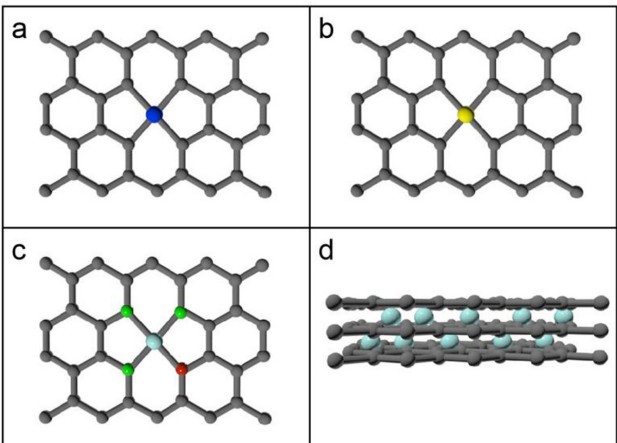

**Figure 1.** The structural schematic diagrams of (**a**) precious metal−doped 2D carbon−based electrocatalysts, (**b**) non−precious metal−doped 2D carbon−based electrocatalysts, (**c**) non-metallic 2D carbon−based electrocatalysts, and (**d**) 2D carbon−based confined electrocatalysts. Blue ball: precious metal dopants; Yellow ball: non−precious metal dopants; Green and red balls: non−metallic dopants; Cyan ball: active centers.

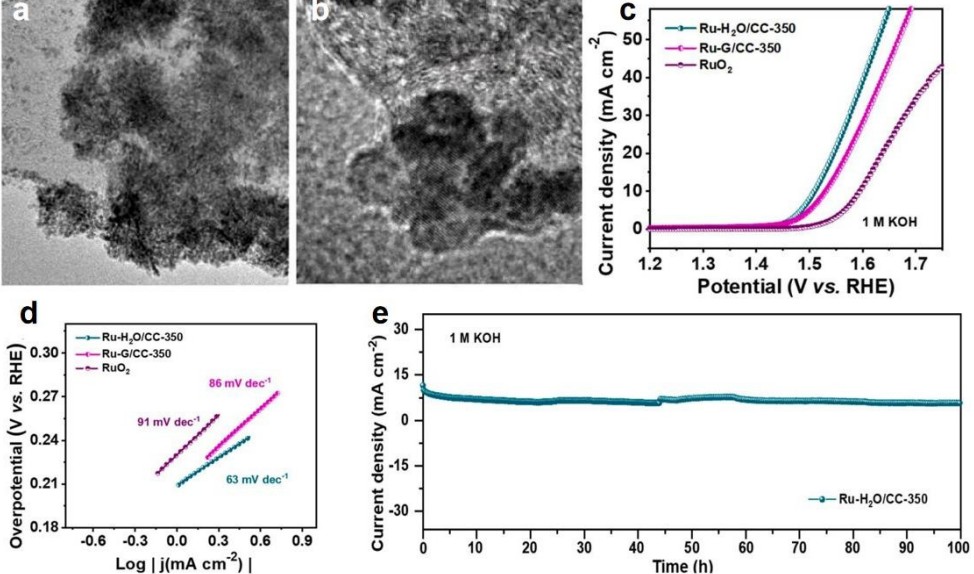

**Figure 2.** (**a**) TEM and (**b**) HR−TEM images of Ru−G/CC. (**c**) LSV curves, (**d**) Tafel plots, and (**e**) chronopotentiometry test of Ru−H$_2$O/CC−350. Reproduced with permission from Ref. [43]. Copyright 2022, Elsevier.

An efficient method to regulate the electronic properties and improve the intrinsic electrocatalytic performance of transition metal−based catalysts is to dope with precious metal and heteroatoms [44–46]. Cobalt−based materials are widely used in electrocatalytic water splitting due to their abundant reserves and low cost [47–51]. However, their electrocatalytic performances are inferior to precious metal−based electrocatalysts due to their poor conductivity and low activity and stability [18,52]. Wang et al. reported that the coupling effect between Ru and Co can improve the catalytic activity because the unusual morphology of the synthesized catalyst shows abundant active sites [53]. Through a one−step pyrolysis procedure and low−temperature oxidation method, the as−synthesized Co−Ru@RuO$_x$/NCN with a core−shell structure possesses the lowest overpotential of 270 mV at the current density of 10 mA cm$^{-2}$ in alkaline solution as compared to Ru@RuO$_x$/NCN (310 mV), NCN (546 mV), and Co$_3$O$_4$/NCN (550 mV) (Figure 3a). Additionally, the Co−Ru@RuO$_x$/NCN shows the smallest Tafel slope of 67 mV dec$^{-1}$ with excellent stability compared with other samples (Figure 3b). Especially, as shown in Figure 3c, Co−Ru@RuO$_x$/NCN only needs the overpotentials of 230 mV and 300 mV to acquire 10 mA cm$^{-2}$ and 50 mA cm$^{-2}$ in acid solution, respectively, which are lower than those of NCN (520 mV and 560 mV, respectively), Co$_3$O$_4$/NCN (340 and 390 mV), and Ru@RuO$_x$/NCN (320 and 410 mV). The Tafel slope value of Co−Ru@RuO$_x$/NCN (48 mV dec$^{-1}$) is the lowest compared to other samples, which further confirms its rapid electrocatalytic kinetics (Figure 3d). Moreover, a decreasing potential of only 12 mV after 10,000 cycles and barely changing for 12 h prove the excellent stability of Co−Ru@RuO$_x$/NCN. The 2D morphology of the CoRu alloy provides sufficient active sites, and the synergistic effect between Co and Ru ensures good electrocatalytic activities [53].

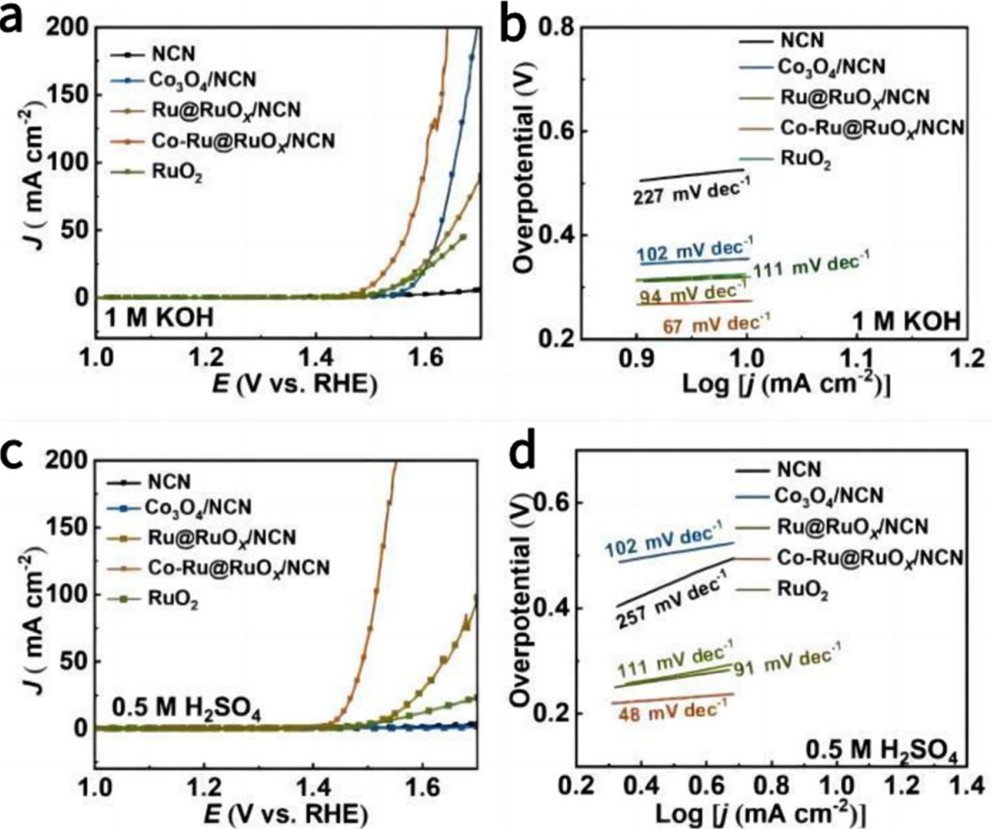

**Figure 3.** (**a**) LSV curves and (**b**) Tafel plots of NCN, Co$_3$O$_4$/NCN, Ru@RuO$_x$/NCN, Co−Ru@RuO$_x$/NCN, and RuO$_2$ to conduct OER in 1.0 M KOH. (**c**) LSV curves and (**d**) Tafel plots of NCN, Co$_3$O$_4$/NCN, Ru@RuO$_x$/NCN, Co−Ru@RuO$_x$/NCN, and RuO$_2$ to conduct OER in 0.5 M H$_2$SO$_4$. Reproduced with permission from Ref. [53]. Copyright 2023, Elsevier.

Because the Ru modulation effects may be applied to other analogous materials, a new method for designing the zeolitic imidazolate frameworks derivatives (ZIF−derived) tri−functional electrocatalysts was developed. Zhang et al. employed hexamethylenetetramine (HMT)−based MOFs as precursors to prepare core−shell Co−Ru nanocomposites with N−doped carbon matrix (CoRu@NC) implanted using a simple pyrolysis process. The $CoRuO_x$@C was obtained by further peroxidation with air to synthesize bimetallic oxide parceled in carbon by one−step pyrolysis [54]. The $CoRuO_x$@C shows better OER activities than those of CoO@C and $RuO_2$@C in alkaline, acid, and neutral solutions. In detail, the $CoRuO_x$@C attains the overpotentials of 240 mV and 223 mV at 10 mA $cm^{-2}$ in alkaline and acidic solutions, respectively (Figure 4a). Meanwhile, the Tafel slopes of $CoRuO_x$@C KOH, PBS, and $H_2SO_4$ are 61.8 mV $dec^{-1}$, 92.2 mV $dec^{-1}$, and 45.0 mV $dec^{-1}$, respectively, which are the lowest compared to CoO@C, $RuO_2$@C, and $RuO_2$ (72.7−132.3 mV $dec^{-1}$). The outstanding electrocatalytic performances are due to the synergistic effect of Co and Ru, the abundant pores of the carbon matrix, as well as the junction of CoRu composites with the carbon matrix [54].

Rh, as a rare precious metal, is about three times more expensive than other precious metals such as Pt and Ru, which severely hinders its utilization for electrocatalytic water splitting [55,56]. In 2020, Sathe's group reported a strategy to integrate Rh nanospheres with conductive graphene oxide (GO) and produced Rh−GO with a face−centered cubic structure to conduct OER [57]. Low−cost GO combined with a tiny amount of Rh equili−brates the price and electrocatalytic performance. The as−prepared Rh−GO only needs the overpotential of 170 mv to achieve the current density of 10 mA $cm^{-2}$ for OER in 0.5 M KOH, which is much lower than that of functionalized GO (470 mV). Moreover, as shown in Figure 4b, the lower Tafel slope of 27 mV $dec^{-1}$ for Rh−GO verifies faster OER kinetics as compared to GO (48 mV $dec^{-1}$). The good durability of the Rh−GO catalyst to conduct OER is further confirmed through an immobility test in Figure 4c, which shows an almost unchanged current density of 10 mA $cm^{-2}$ at the potential of 1.4 V versus the RHE during OER. The inexpensive GO and Rh nanospheres provide high porosity and active surface area, which ensure excellent OER performances [57].

Due to high activity and excellent corrosion resistance in acidic medium, iridium (Ir)−based materials, such as $IrO_2$, metallic Ir, and $IrO_x$, are regarded as the benchmark for OER electrocatalysts [58−61]. However, in the current study, the morphologies of the Ir−based catalysts are mostly the self−assembly of spherical and cylindrical mi−celles. There are few reports that Ir−based catalysts have 2D structures deriving from the self−assembly of lamellar micelles. In this work, a nanoconfined self−assembly strat−egy via stable end−merge lamellar micelles to prepare novel 2D nanomaterials that have ordered mesoporous interlayer spaces was shown by Zu et al (Figure 4d) [62]. When the current density is 10 mA $cm^{-2}$, the as−prepared mesoporous Ir−$IrO_x$/C−20 has the lowest overpotential of 198 mV (Figure 4e), compared with Ir−$IrO_x$/C−20, Ir−$IrO_x$/C−30, Ir/C, Ir−$IrO_x$/C−10, and $IrO_2$. Additionally, the onset potentials of Ir−$IrO_x$/C−10, Ir−$IrO_x$/C−20, and Ir−$IrO_x$/C−30 are 1.35, 1.37, and 1.38 V, respectively. According to the experiments, the Tafel slope of Ir−$IrO_x$/C−20, Ir−$IrO_x$/C−10, Ir−$IrO_x$/C−30, and com−mercial $IrO_2$ are, respectively, 106.3 mV $dec^{-1}$, 257.2 mV $dec^{-1}$, and 115.1 mV $dec^{-1}$, indi−cating that Ir−$IrO_x$/C−20 has the best OER kinetics among Ir−$IrO_x$/C catalysts (Figure 4f). The metallic Ir0 nanocrystal core can boost the adsorption energy of oxygen−containing species, and $IrO_x$ can reduce the adsorption free energy of *OOH, which effectively balances the interaction between OER and oxygen intermediates, proving that the mixed−valence catalyst is conducive to optimizing the adsorption energy of OER to oxygen−containing species [62].

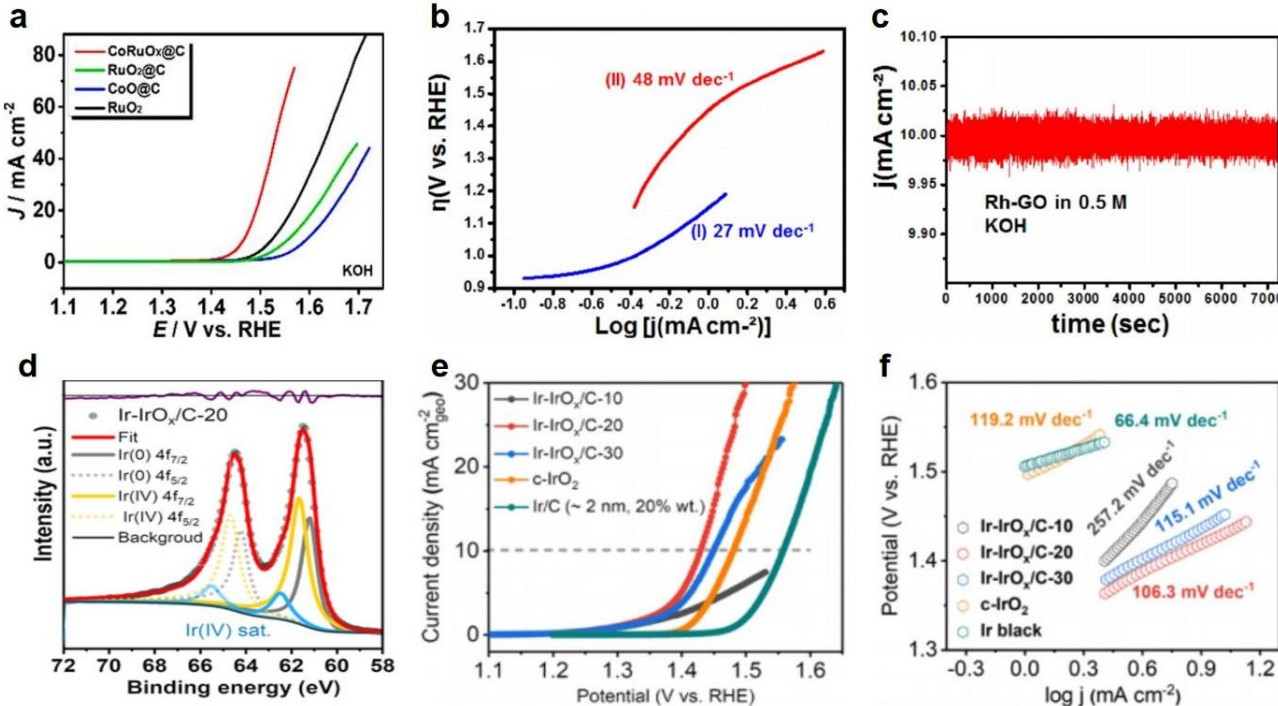

**Figure 4.** (**a**) LSV curves of CoRuO$_x$@C and other contrast samples in 1.0 M KOH. Reproduced with permission from Ref. [54]. Copyright 2022, Elsevier. (**b**) Tafel plots of Rh−GO (27 mV dec$^{-1}$) and counterpart GO (48 mV dec$^{-1}$). (**c**) Immobility test of Rh−GO to conduct OER in 0.5 M KOH. Reproduced with permission from Ref. [57]. Copyright 2020, American Chemical Society. (**d**) Nitrogen adsorption−desorption isotherm and the corresponding pore-size distribution curves (inset) of the Ir−IrOx/C-20. (**e**) LSV curves and (**f**) Tafel plots of Ir−IrO$_x$/C−20 and other contrast samples. Reproduced with permission from Ref. [62]. Copyright 2022, American Chemical Society.

The examples we summarized above use the nanoconfined self−assembly strategy, pyrolysis method, low−temperature oxidation process, and other methods to synthesize materials. Precious metals supported on conductive materials such as carbon are common electrocatalysts for a wide range of electrochemical reactions. Ir, Ru, Rh, and other precious metal materials are embedded with carbon materials, and the precious metal material serves as the active site (Table 1). Both the high catalytic activity and stability of precious metals and the synergistic effect between precious metals and carbon-based materials are conducive to the improvement in the activity of electrocatalysts [63].

**Table 1.** Precious metal doped 2D carbon−based electrocatalysts.

| Catalyst | Carbon Precursor | Precious Metal | Electrolyte | Overpotential (mV@10 mA cm$^{-2}$) | Tafel Slope (mV dec$^{-1}$) | Surface Area (m$^2$ g$^{-1}$) | Stability | Ref. |
|---|---|---|---|---|---|---|---|---|
| Ru@g−CN$_x$ | graphitic carbon nitride | Ru | 1 M KOH | 280 | 49.5 | 23 | 45 h | [42] |
| Ru−H$_2$O/CC−350 | CC substrate | Ru | 1 M KOH | 270 | 63 | — | 100 h | [43] |
| Co−Ru@RuO$_x$/NCN | NCN | Ru | 1 M KOH | 270 | 48 | 603.45 | 12 h | [53] |
| Ru@RuO$_x$/NCN | NCN | Ru | 1 M KOH | 310 | — | — | — | [53] |
| CoRuO$_x$@C | N−doped carbon matrix | Ru | 1 M KOH | 240 | 61.8 | — | — | [54] |
| Rh−GO | GO | Rh | 0.5 M KOH | 230 | 27 | 8.909 | — | [57] |
| Ir−IrO$_x$/C−20 | C−20 | Ir−IrO$_x$ | 0.5 M H$_2$SO$_4$ | 198 | 106.3 | 146 | 18 | [62] |
| Ir−IrO$_x$/C−10 | C−10 | Ir−IrO$_x$ | 0.5 M H$_2$SO$_4$ | — | 257.2 | 182 | — | [62] |
| Ir−IrO$_x$/C−30 | C−30 | Ir−IrO$_x$ | 0.5 M H$_2$SO$_4$ | — | 115.1 | 114 | — | [62] |

### 3. Non−Precious Metal−Doped 2D Carbon−Based Electrocatalysts

Nowadays, the state−of−the−art commercial catalysts for OER are still Ru/Ir−based oxide materials [64,65]. Nevertheless, the scarcity and high cost of these precious metals have dramatically impeded their large−scale applications. One of the strategies to solve the bottleneck is to develop non−precious metal−doped 2D carbon−based electrocatalysts as alternatives [15]. The structure of non−precious metal−doped 2D carbon−based electrocatalysts is shown in the Figure 1b. Next, we will introduce several relevant research achievements, such as Fe−NG, FeCo/NB−Cs, Ni−Co−P/GDY, Rh@R−graphyne, and Ni@R−graphyne.

Because of characteristics such as a large specific surface, plasticity, and high conductivity, 2D graphene is widely used in energy storage−related fields as the cornerstone for constructing carbon−based electrocatalysts [66–68]. Additionally, the synergistic effect produced by doping the non−precious metals accelerates the redistribution of positive and negative charges in graphene, leading to the enhancement of the conductivity and charge transfer of the whole system, as well as the relevant electrocatalytic activity [69,70] Due to the strong binding affinity between Fe and oxygen and the synergistic effect of the Fe−$N_x$ bond in Fe/NG, iron and nitrogen co−doped graphene−like (Fe/NG) materials have been widely studied in recent years as excellent bifunctional electrocatalysts for OER because of the large specific surface area, abundant exposed active sites, and high nitrogen content [71,72]. An N−containing polymer (2,5−benzimidazole) (ABPBI) and iron precursor were inserted into CMMT for pyrolysis to prepare 2D non−metal N−doped graphene (2D NG) and bi−functional iron/nitrogen co−doped graphene (2D Fe−NG) electrocatalyst by Wang et al. [73]. Because of the limit of the layer template for the precursors, the 2D graphene and 2D Fe−NG have a high BET surface area and 2D graphene−like structure. The OER LSV curves show that at a current density of 10 mA cm$^{-2}$, the overpotential of 2D Fe−NG is 390 mV in 0.1 M KOH electrolyte (Figure 5a), which is better than 2D NG (403 mV) and slightly higher than $RuO_2$ (370 mV), and the Tafel slope of 2D Fe−NG (70.1 mV dec$^{-1}$) is higher than $RuO_2$ (67.9 mV dec$^{-1}$) and closer to 2D NG (71.3 mV dec$^{-1}$) (Figure 5b). Using the i−t chronoamperometry method, the OER durability of 2D Fe−NG, 2D NG, and $RuO_2$ was evaluated, and the current density of 2D Fe−NG has a slight loss of 13.5% after 50,000 s, better than the 19.3% of 2D NG and 33% of $RuO_2$. The low overpotential and stable OER durability indicate that 2D Fe−NG has excellent OER performance. The addition of iron and novel ABPBI precursors that are rich in nitrogen promotes the formation of OER active sites. The doping of Fe and N in the Fe/NG forms a Fe−$N_x$ bond which produces a synergistic effect, accelerating the formation process of OER active sites.

Known as serious prospects and outstanding electrocatalysts, heterogeneous doped 2D carbon including non−precious metal atoms have the ability to lower the overpotential of OER. Taking existing research and challenges into consideration, carbon materials doped with bi−nonmetal (N/B) and bi-metal dopants, providing abundant active sites, could be employed as excellent catalysts for water oxidation. The Fe/NB−Cs and FeCo/NB−Cs Li et al. prepared perform micro/mesoporous structure [74]. Figure 5c,d show that the overpotentials to obtain the current density of 10 mA cm$^{-2}$ of Fe/N−Cs, Fe/NB−Cs and FeCo/NB−Cs are 328 mV, 320 mV, and 271 mV, respectively, implying the superior OER electrocatalytic performances of FeCo/NB−Cs, which transcends those of Pt/C (682 mV) and $RuO_2$ (343 mV). Furthermore, the overpotential of FeCo/N−Cs at 10 mA cm$^{-2}$ is 292 mV, which suggests the Co dopant could effectively boost the OER electrocatalytic activity. The density functional theory (DFT) calculation uncovers that the sensible synergetic effect between Fe/Co and N/B dopants boosts the OER catalytic activities. The legitimate strategy to construct heterogenous−doped 2D carbon−based materials could synergetically contribute diverse active sites. Graphdiyne (GDY) has attractive properties such as a heavily exposed surface, conductive carbon backbone, and high robustness and is, therefore, considered an appropriate support material [75–78]. In general, as an OER electrocatalyst, Ni−Co−P/GDY with 2D/2D heterojunction manifests excellent performance under alkaline conditions, owing to the synergistic effect of Ni−Co−P and GDY [79].

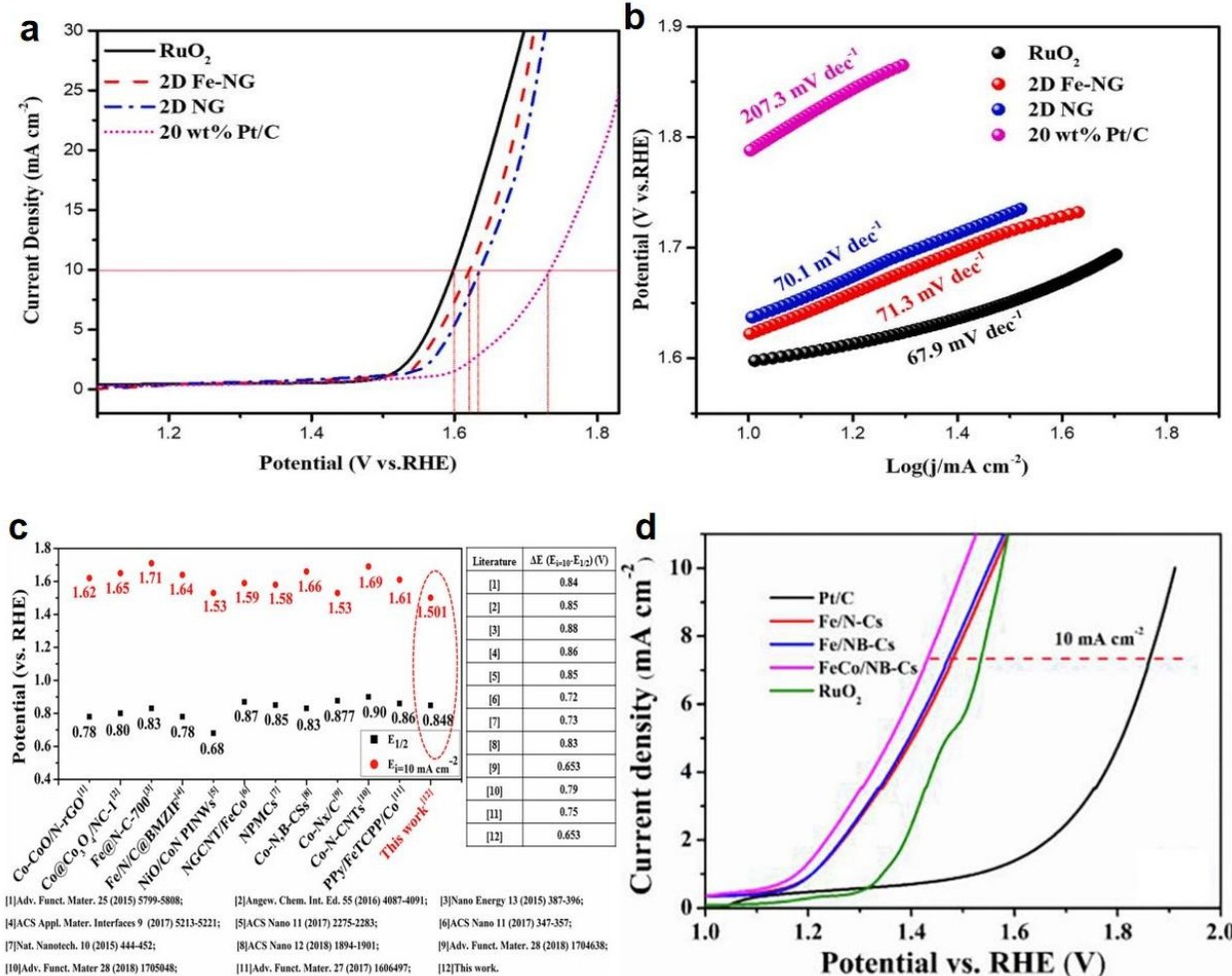

**Figure 5.** (**a**) LSV curves and (**b**) Tafel plots of Fe−NG and other contrast samples. (**c**) Comparison of the required voltage at 10 mA cm$^{-2}$ for FeCo/NB−Cs with other bifunctional catalysts. (**d**) LSV curves of FeCo/NB−Cs and other contrast samples in 1.0 M KOH. Reproduced with permission from Ref. [74]. Copyright 2019, Elsevier.

As a promising 2D carbon allotrope composed of tetra−rings and acetylenic link-ages, rectangular graphyne (R−graphyne) is promising for renewable energy conversion owing to its thermodynamic stability and unique electronic properties. Developing 2D nanomaterials formed by R−graphyne with other materials can afford a new route to realize high−performance and low−cost electrocatalysts for OER [80]. All structural units of R−graphyne have anti−aromaticity, which can make the relevant carbon atoms have excellent reactivity. Li et al. proposed an effective approach for improving the OER catalytic activity by doping Group VIIIB elements with 2D R−graphyne [15]. The Rh@R−graphyne and Ni@R−graphyne with low−dimensional nanostructures display good electrocatalytic OER performances of lower calculated overpotentials of 0.48 V and 0.31 V in contrast to those of Ru@R−graphyne (1.43 V), Ir@R−graphyne (1.16 V), and Co@R−graphyne (0.74 V). Evidently, implanting Ni and Rh atoms in Group VIIIB can greatly boost the OER catalytic performance of R−graphyne. The good OER performances could be attributed to the synergistic effect between metal dopants and R−graphyne, as well as increased antibonding characteristics that offer a proper adsorption state of O* [15].

The above examples summarize the salt template method, molten−salt−assisted pyrolysis method, pyrolysis method, and many other methods to synthesize materials. Non−precious metal carbide−based carbon composites are formed by filling the void of the non−precious metal lattice with carbon atoms. Non−precious metal elements such

as nickel, cobalt, iron, copper, and manganese at the edge position are considered to be the main reaction centers in non−precious metal composites. Thanks to the modification of intermediate binding, OER at marginal sites has lower overpotential. Due to the large quantities of non−precious metals on Earth, they have attracted great attention for use as catalysts. The coupling effect of non−precious metals and carbon−based materials promotes better charge transfer and increases the activity of the catalyst [81].

## 4. Non−Metallic 2D Carbon−Based Electrocatalysts

As discussed above, graphene hybrid materials have the characteristic of high electrical conductivity, which promotes the electrochemical process. The structure of non−metallic 2D carbon−based electrocatalysts is shown in Figure 1c. Non−metallic carbon catalysts have become a promising research object, and next, we will introduce several relevant research achievements, such as EBP@NG, g−$C_3N_4$/rGO, F/BCN, and CNS-0.5N.

Heteroatom−doped carbon materials, especially nitrogen−doped carbon materials, are promising materials as metal−free OER catalysts because the charge distribution and electronic structure of nearby carbon are effectively regulated by N atoms, improving catalytic activity [82,83]. Graphene has great potential as support for N−doped carbon assembly owing to its special 2D single−atom−thick π−conjugated structure, high surface area, and excellent electrical conductivity [84]. The few−layered exfoliated black phosphorus (EBP) nanosheet has the features of high carrier mobility, tunable electronic structure, large specific surface area, and full scalability. Due to easy oxidation in air, the EBP is usually coupled to graphene−based materials to obtain electrocatalysts for water oxidation [85]. Yuan et al. constructed a novel metal−free 2D/2D heterostructure via electrostatic interaction of positively charged N−doped graphene (NG) and negatively charged EBP, which was denoted as EBP@NG [86]. The OER performances of EBP@NG and other contrast samples were tested in 1.0 M KOH. As shown in Figure 6a,b, the optimized EBP@NG (1:8) shows the lowest overpotential of 310 mV at 10 mA $cm^{-2}$ as compared to those of common EBP (>500 mV) and NG (430 mV), and is even comparable to the commercial $RuO_2$ (300 mV). Moreover, the EBP@NG (1:8) displays a Tafel slope of 89 mV $dec^{-1}$ as compared to the $RuO_2$ catalyst (78 mV $dec^{-1}$) which affords decent OER kinetics. Specially, the current loss of EBP@NG (1:8) is only less than 4% after an operation at the current density of 10 mA $cm^{-2}$, while the current losses of bare EBP and $RuO_2$ are larger than 50% after 2 h and 50% after 5 h, respectively. The DFT calculations and experimental results imply that the synergistic effect between EBP and NG optimizes the adsorption energies of OER intermediates, which promote the formation of OOH* and finally improve the OER energetics [86].

The as−prepared g−$C_3N_4$/rGO displays an onset potential of 1.55 V, which is lower than those of rare rGO (1.58 V) and g−$C_3N_4$ (1.64 V), implying smaller intrinsic resistance and efficient active sites. Further, the 2D g−$C_3N_4$/rGO attains the lowest overpotential of 272 mV at the current density of 10 mA $cm^{-2}$ as compared to those of rGO and g$C_3N_4$, which are 317 mV and 420 mV, respectively (Figure 6c). Furthermore, as shown in Figure 6d, the g−$C_3N_4$/rGO displays the lowest Tafel slope of 97 mV $dec^{-1}$ as compared to rGO (127 mV $dec^{-1}$) and g−$C_3N_4$ (266 mV $dec^{-1}$). According to the comparison stability tests of 2D heterogeneous g−$C_3N_4$/rGO and commercial $RuO_2$, the g−$C_3N_4$/rGO shows the ignored value change of current density for 24 h at the potential of 1.5 V. The excellent electrocatalytic performances of the carbon−based heterogeneous g−$C_3N_4$/rGO with 2D/2D heterostructure could be attributed to the introduction of 2D materials, which enhances the electron transfer to the interface between electrodes and optimizes the electrocatalytic active sites for OER [87].

Although boron carbon nitride nanosheets (BCN NSs) have been developed in electrocatalysis because of the idealized physical and physicochemical properties of the two dimensions of both 2D hexagonal boron nitride (h−BN) and graphene [88–90], the performance of BCN NSs in OER process did not meet expectations, which limits their further developments in electrocatalysis [91]. Buckminsterfullerene (C60), which is a 0D carbon

structure, is regarded as a candidate for building efficient multifunctional metal−free hybrid electrocatalysts due to superior electron−accepting properties and high affinity for constructing supra molecular assemblies. In their work, Md Ariful Ahsan and his group mixed prepared 50 mg BCN NSs powder with 20 mL isopropyl alcohol (IPA) in a beaker and treated the mixture of fullerene solution in toluene (from 5 to 30 wt%) and BCN NSs solution with bath sound for 30 min to transform F/BCN nanohybrids. F/BCN was prepared by washing F/BCN nanohybrids with toluene and water several times and drying them at 70 °C overnight in a vacuum oven. It was calculated that 10% F/BCN requires 390 mV to achieve 10 mA cm$^{-2}$ current density, while the benchmark RuO$_2$ catalyst requires 410 mV (Figure 6e). The Tafel slopes can reflect the catalytic activity and kinetics of catalysts, and the Tafel slope of 10% F/BCN is the lowest among all samples, which is 79 mV dec$^{-1}$ (Figure 6f), indicating that it has superior OER kinetics on the catalytic surface [92]. Therefore, the existence of a synergistic arrangement that forms a supramolecular interface with catalytic properties of the metal−free 10% F/BCN nanohybrid material was confirmed, indicating that the electrochemical stability of F/BCN nanohybrid materials is better than that of existing commercial metal−based catalysts.

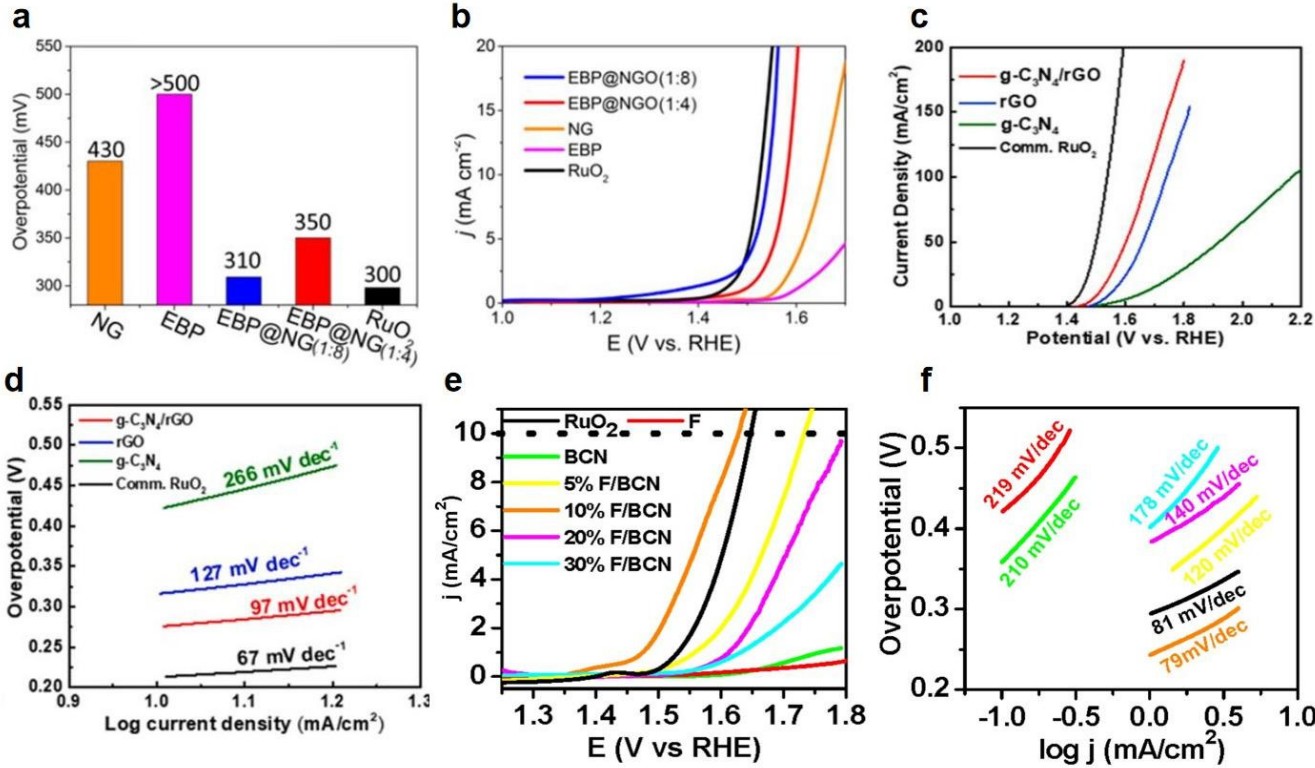

**Figure 6.** (**a**) Overpotentials and at 10 mA cm$^{-2}$ of EBP@NG and other contrast samples in 1.0 M KOH. (**b**) LSV curves of EBP@NG and other contrast samples in 1.0 M KOH to conduct OER. Reproduced with permission from Ref. [86]. Copyright 2019, American Chemical Society. (**c**) LSV curves and (**d**) Tafel plots of g−C$_3$N$_4$/rGO and other contrast samples. Reproduced with permission from Ref. [87]. Copyright 2022, Elsevier. (**e**) LSV curves and (**f**) Tafel plots of F/BCN and other contrast samples in 0.5 M NaOH to conduct OER. Reproduced with permission from Ref. [92]. Copyright 2021, American Chemical Society.

Graphitic carbon nitride could be employed as a 2D metal−free catalyst with moderate catalytic activity. In recent years, various strategies have been practiced to enhance the OER electrocatalytic performances of carbon nitride [93–97]. Although there are some achievements, exploring simple and economical ways to modify CN with more active sites remains a considerable challenge. An acid−induced method was employed by Huang's group [96] to develop a homojunction of S−doped graphitic carbon nitride with graphitic carbon ni-

tride through easily pyrolyzing a supramolecular precursor. The optimized 2D CNS−0.5N gives the lowest onset potential of 1.48 V as compared to other samples (1.49−1.53 V), which verifies its rapid increase of current density and good OER performance in 1.0 M KOH. Further, the overpotential of CNS−0.5N is 301 mV at the current density of 10 mA cm$^{-2}$, which is superior to most non−metallic electrocatalysts and the commercial IrO$_2$ (>360 mV) [98]. The quick electron transfer and the reaction acceleration of CNS−0.5N are confirmed by the low Tafel slope of 57.71 mV dec$^{-1}$. The excellent OER performances could be ascribed to the sufficient active sites supported by the large surface area and effective charge transfer and separation provided by the 2D S−CN/CN homojunction [96].

The electrostatic self−assembly method, in situ self−assembly, the hydrothermal method, and many other methods are used in the abovementioned examples. Introducing B, N, S, and other non−metallic heteroatoms into the carbon skeleton may produce more defects, which can improve the activity of the catalyst. Due to the existence of defects and heteroatoms, the non−metallic atoms have abundant active centers, more efficient charge transfer, large specific surface area, and high conductivity, which makes the non−metallic 2D carbon−based electrocatalysts have an OER performance (Table 2). Non−metallic catalysts are considered ideal commercial catalysts because of their large surface area, excellent electrical conductivity and appropriate cost, and the synergistic effect between multiple atoms can further improve the catalytic capacity of catalysts [99,100].

**Table 2.** Non−precious metal doped and non−metallic 2D carbon−based electrocatalysts.

| Catalyst | Carbon Precursor | Doping Material | Electrolyte | Overpotential (mV@10 mA cm$^{-2}$) | Tafel Slope (mV dec$^{-1}$) | Surface Area (m$^2$ g$^{-1}$) | Stability | Ref. |
|---|---|---|---|---|---|---|---|---|
| Ni@R−graphyne | R-graphyne | Ni | — | 310 | — | — | — | [15] |
| Fe−NG | graphene | Fe, N | 0.1 M KOH | 390 | 70.1 | 714.5097 | 80h | [73] |
| NG | graphene | N | 0.1 M KOH | 403 | 71.3 | 563.7250 | — | [73] |
| FeCo/NB−Cs | mesoporous carbon nanosheets | Fe−Co and N−B | 0.1 M KOH | 653 | — | 1584 | 6700 min | [74] |
| Fe/N−Cs | mesoporous carbon nanosheets | Fe | 0.1 M KOH | 745 | — | 1235 | — | [74] |
| Fe/NB−Cs | mesoporous carbon nanosheets | Fe | 0.1 M KOH | 729 | — | 1654 | — | [74] |
| Ni−Co−P/GDY | GDY | Ni−Co−P nanosheets | 1 M KOH | 290 | 72.7 | — | 45 h | [79] |
| EBP@NG(1:4) | NG | EBP | 1 M KOH | 350 | 82 | — | 16 h | [86] |
| EBP@NG(1:8) | NG | EBP | 1 M KOH | 310 | 89 | — | — | [86] |
| g−C$_3$N$_4$/rGO | rGO | g−C$_3$N$_4$ | 1 M KOH | 272 | 97 | 142.49 | 24 h | [87] |
| 10% F/BCN | CN | F, B | 0.5 M NaOH | 390 | 79 | — | 12 h | [92] |
| CNS−0.5N | CN | S | 1 M KOH | 301 | 57.71 | — | — | [96] |

## 5. 2D Carbon−Based Confined Electrocatalysts

As known, 2D layered materials are outstanding substrates for OER electrocatalysts, which could not only decrease the metal input by increasing the surface area for active sites but also enhance the stability during OER through the tight combination of metals to carbon matrix [101,102]. In particular, 2D carbon−based materials could be employed to construct confinement environments for electrocatalysts in order to provide excellent OER activity and stability [103,104]. The structure of 2D carbon−based confined electrocatalysts is shown in the Figure 1d. Next, we will introduce several relevant research achievements, such as NiFe−BTC//G, FeNi@NCSs, and Co$_3$O$_4$@NCNs.

Metal−organic frameworks (MOFs) are a series of promising materials to conduct OER due to their large surface area, adjustable porosity, tunable compositions, and metal centers. However, the intrinsically bad electroconductivity and poor stability of MOFs severely hamper their application for water oxidation, which needs to be solved [105–107]. In 2022, Lyu et al. innovatively confined a bimetallic NiFe−based MOF into 2D graphene multi-

layers to obtain 2D NiFe−BTC//G through a universal strategy of simple electrochemical intercalation [108]. The as-prepared NiFe−BTC//G displays a record low overpotential of 106 mV at the current density of 10 mA cm$^{-2}$ in 1.0 M KOH to conduct OER, which exceeds all other MOF−based electrocatalysts (Figure 7a). In contrast, the monometallic Ni−BTC//G and Fe−BTC//G, the bulk NiFe−BTC, as well as the commercial RuO$_2$ and Ir/C, show higher overpotentials at 10 mA cm$^{-2}$ as compared to NiFe−BTC//G, which are 212 mV, 226 mV, 399 mV, 267 mV, and 287 mV, respectively. Furthermore, the novel NiFe−BTC//G displays the lowest Tafel slope of 55 mV dec$^{-1}$ compared with the bulk NiFe−BTC (189 mV dec$^{-1}$), Ir/C (76 mV dec$^{-1}$), and RuO$_2$ (103 mV dec$^{-1}$), showing the most favorable electrocatalytic OER kinetic (Figure 7b). Significantly, the NiFe−BTC//G possesses an ignored decline of potential at the current density of 10 mA cm$^{-2}$ for 150 h, which confirms an outstanding electrocatalytic stability to conduct OER. The nanoconfinement offered by graphene multilayers ensures the formation of highly active species and, thus, greatly enhances the electrocatalytic performances of OER [108].

A bimetallic FeNi alloy was confined to N−doped carbon nanosheets through a simple complexation pyrolysis strategy by Lin et al. to catalyze water oxidation (Figure 7c,d) [109]. The as−developed FeNi@NCSs electrocatalyst only needs a lower overpotential of 397 mV in the O$_2$−saturated 1.0 M KOH compared with FeNi−900 (405 mV), FeNi−700 (485 mV), and the commercial RuO$_2$ (432 mV) to achieve a current density of 100 mA cm$^{-2}$. Furthermore, the FeNi@NCSs display the lowest Tafel slope of 40.8 mV dec$^{-1}$ as compared to other contrast samples, which confirms the good electrocatalytic OER dynamics. The outstanding electrocatalytic OER performances could be attributed to the ample active sites and high graphitic degree. Importantly, the confinement environment provided by N−doped carbon nanosheets makes it easier for active sites to approach the electrolyte [109].

N−contained precursors, such as porous organic molecules and MOFs, are usually used to prepare novel carbon−based materials by pyrolysis to conduct OER [110,111]. However, the loss of carbon and N owing to the high pyrolysis temperature results in the low productivity of carbon−based materials with a low N loading amount. One−step pyrolysis of NaCl−encapsulated ZnO@zeolitic imidazolate framework nanoparticles was used by Xi et al. to prepare defect−rich N−doped carbon nanosheets on Co$_3$O$_4$ [112]. The 2D confined Co$_3$O$_4$@NCNs electrocatalyst shows the overpotential of 240 mV at the current density of 10 mA cm$^{-2}$ as compared to the counterparts of Pt/C+RuO$_2$ (270 mV) and Co−NC (470 mV) (Figure 7e). As shown in Figure 7f, the as−developed Co$_3$O$_4$@NCNs gives the lowest Tafel slope of 90 mV dec$^{-1}$ compared with the counterpart Co−NC (220 mV dec$^{-1}$) and the commercial Pt/C+RuO$_2$ (107 mV dec$^{-1}$), confirming the fast electrocatalytic OER kinetics of Co$_3$O$_4$@NCNs. Furthermore, the OER stability of Co$_3$O$_4$@NCNs evaluated by chronopotentiometry shows that only 70 mV of the increased overpotential is observed after 25,000 s. The good electrocatalytic OER performances could be ascribed to the NaCl confinement that hinders the intermediates and produces the holed NCNs [112].

The abovementioned examples used complexation pyrolysis, one−step pyrolysis, and other methods to synthesize the unmeasured electrocatalysts. The confinement of the active sites and adsorbates between 2D carbons leads to the modulation of electronic states. Actually, the favorable interaction between the active centers and the functional 2D carbon layers could help the self−assembly of nanoparticles in the desired confined space. The multipath transfer of ions during the dynamic structural transformation continuously activates the catalytic behavior and results in good OER performances of 2D carbon−based confined electrocatalysts. In the confined catalysts, electrons are transferred from the nanoparticles to the packaged carbon shell or carbon nanotubes, which embellish the electronic structure of the non−activated carbon and improve the electrocatalytic performance significantly [113].

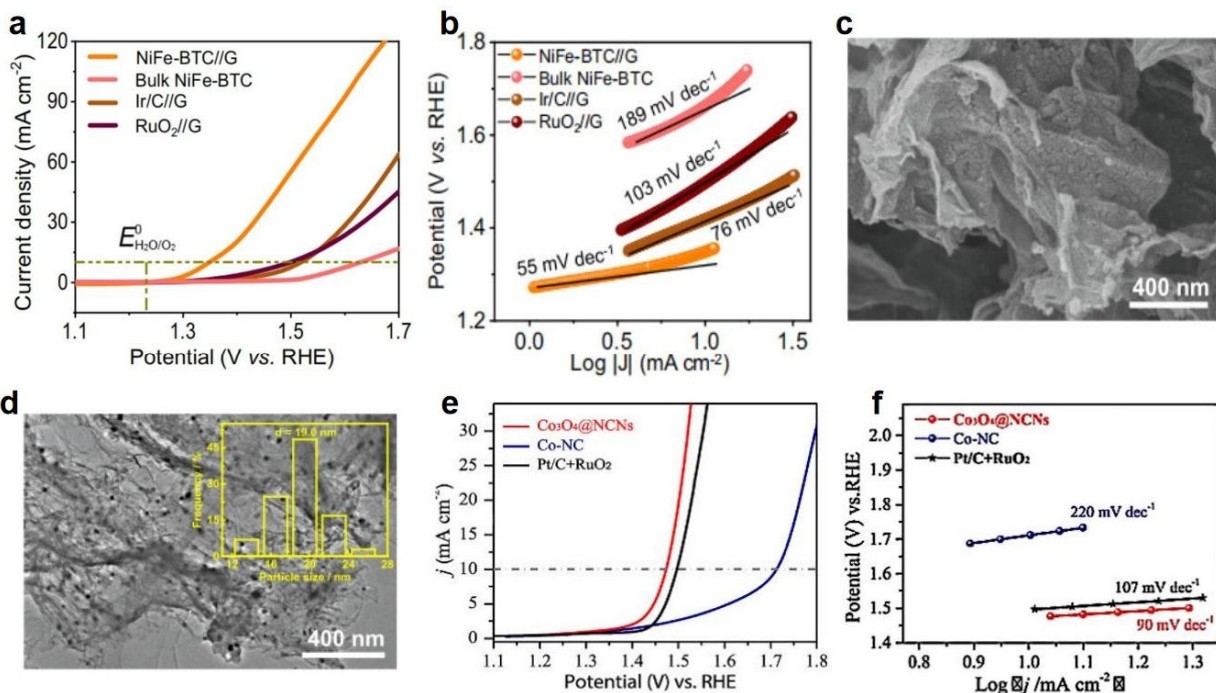

**Figure 7.** (**a**) LSV curves and (**b**) Tafel plots of NiFe−BTC//G and other contrast samples in 1.0 M KOH for OER. Reproduced with permission from Ref. [108]. Copyright 2022, Nature Publishing Group. (**c**) SEM and (**d**) TEM images of FeNi@NCSs. Reproduced with permission from Ref. [109]. Copyright 2022, Elsevier. (**e**) LSV curves and (**f**) Tafel plots of $Co_3O_4$@NCNs and other contrast samples in 0.10 M KOH to conduct OER. Reproduced with permission from Ref. [112]. Copyright 2023, Elsevier.

## 6. Summary and Outlook

The research progress and application of 2D carbon−based materials as OER electrocatalysts are summarized in this review. These 2D carbon−based materials are considered promising OER electrocatalysts because of their large surface area, excellent electrical conductivity, abundant active sites, high porosity, good durability, and low cost [114]. Four classes of recently reported 2D carbon−based electrocatalysts, including precious metal−doped 2D carbon−based electrocatalysts, non−precious metal−doped 2D carbon−based electrocatalysts, non−metallic 2D carbon−based electrocatalysts, and 2D carbon−based confined electrocatalysts are systematized. We also summarize some preparation methods for 2D carbon−based materials, such as pyrolysis, solvothermal, template, hydrothermal, in−situ reduction methods, and so on. Precious metal−based electrocatalysts have excellent OER performance, but their high cost limits their development space, while 2D carbon−based materials doped with precious metals show excellent activity and performance due to the synergistic effect between 2D carbon and precious metals, especially the coupling effect of Co and Ru.

Furthermore, 2D carbon materials doped with the precious metal Rh also show outstanding performances in conducting OER. The interaction between non−precious metal materials and carbon materials in improving catalyst performance is also increasingly studied. The as−developed non−precious metal−doped 2D carbon−based electrocatalysts impose an electron−deficiency site, creating a synergistic effect, reducing the overpotential of the water decomposition and greatly improving OER performance. In this work, the electrocatalysts were prepared by doping non-precious metal materials (Ni, Co, Fe, etc.) and two-dimensional carbon−based materials (graphene, graphene, carbon nanosheets, etc.) in various ways. Moreover, it is known that those electrocatalysts have unique advantages for water oxidation, such as good conductivity, low cost, large specific surface area, high porosity, abundant active sites, and good durability [57]. Combining 2D carbon−based

materials with non−metallic materials (N, S, P, F, etc.) to prepare OER electrocatalysts has attracted much attention because of its high cost effectiveness. The electron interaction between 2D carbon−based and non−metallic materials induces directional interfacial electron transfer, which regulates the adsorption energy of OER intermediates and greatly enhances OER energy. In particular, it is a feasible method to reduce metal input and improve electrocatalytic OER efficiency by using 2D carbon−based materials as substrates in confined catalysts. The nanoconfinement provided by 2D carbon−based materials could shorten the transmission distance of intermediates, lower the limiting potential for water oxidation reaction, and induce the formation of highly active sites, as well [108]. Although 2D carbon−based materials show great potential in OER, there is still a lot of room for developing their electrocatalytic OER performance, especially in challenging acidic electrolytes. The 2D carbon−based materials are susceptible to corrosion, and their durability under working conditions needs to be enhanced. Those insufficiencies will become an obstacle to further development and application. Moreover, the doping amounts of heteroatoms in the catalysts are usually very low, and the doping types are difficult to control [115–118] and the OER electrocatalysts with complex structures are faced with the difficulty of identifying active sites in complicated electrochemical environments [119–123], which may become obstacles to the further improvement of catalytic activity. Furthermore, the properties of individual 2D carbon nanosheets may be affected because they can aggregate, overlap, or restack due to the Van Der Waals attraction between the slices and the high surface energy. On the other hand, the study of highly efficient and low−cost 2D carbon−based materials is conductive to their industrialization. Understanding the reaction mechanism, kinetics, and the relationship between the reaction mechanism and OER performance of 2D carbon-based electrocatalysts is helpful in designing efficient catalysts. Theoretical calculation and advanced characterization techniques such as in situ Raman spectroscopy, in situ Fourier−transform infrared spectroscopy, and in situ X−ray absorption near-edge structure play an important role in developing efficient 2D carbon−based electrocatalysts for OER. In the meantime, it is necessary to continue the research on 2D carbon−based nanomaterials for flexible devices with high mechanical strength and shape conformability, which could achieve utilization in foldable, portable, and wearable energy systems. Moreover, the applications for 2D carbon−based nanomaterials in other practical energy devices, such as rechargeable metal−air batteries, fuel cells, and solar cell devices, can provide more opportunities in related energy sectors. Last but not least, the reaction descriptors for OER should be further developed to predict specific sets of electrocatalysts, explain the fundamental OER facets, and achieve high−throughput computational screening for hypothetically high-performing catalysts.

In summary, this review summarizes the application and development of 2D carbon−based materials as OER electrocatalysts in recent years, which is helpful in promoting theoretical research and technological innovation in related academic fields. We introduce the 2D carbon−based materials from four aspects: precious metal−doped 2D carbon−based electrocatalysts, non−precious metal−doped 2D carbon−based electrocatalysts, non−metallic 2D carbon-based electrocatalysts, and 2D carbon−based confined electrocatalysts. Understanding the preparation, mechanisms, and related properties of OER electrocatalysts is conductive to revolutionizing the future energy system and reducing harmful gas emissions and dependence on petroleum products.

**Author Contributions:** S.L., C.L. and T.W. conceived the idea and supervised the project. Y.Z. and S.N. wrote the review article. B.X. searched the recently important research work. Z.D. and T.Y. helped to revise the manuscript. All authors have read and agreed to the published version of the manuscript.

**Funding:** This research was funded by the Doctoral Scientific Research Foundation of Hubei University of Automotive Technology, the National Natural Science Foundation of China (Grant No. 22208076), Zhejiang Provincial Natural Science Foundation of China (Grant No. LQ23B060001), start−up funding from Hangzhou Normal University (Pandeng II Plan Foundation: 2021QDL068).

**Data Availability Statement:** The data presented in this study are available on request from the corresponding author.

**Conflicts of Interest:** The authors declare no competing interests.

## Abbreviations

| Abbreviation | Full Name |
| --- | --- |
| HER | hydrogen evolution reaction |
| OER | oxygen evolution reaction |
| 2D | two−dimensional |
| VCNs | vertically aligned carbon nanosheets |
| Ru | ruthenium |
| ZIF-derived | zeolitic imidazolate frameworks derivatives |
| HMT | hexamethylenetetramine |
| GO | graphene oxide |
| Ir | iridium |
| N/B | bi−nonmetal |
| DFT | density functional theory |
| GDY | graphdiyne |
| CPT | cathodic polarization treatment |
| CVD | chemical vapor deposition |
| ABPBI | 2,5−benzimidazole |
| IPA | isopropyl alcohol |
| MOFs | metal−organic frameworks |
| C60 | buckminsterfullerene |
| h-BN | hexagonal boron nitride |
| BCN NSs | boron carbon nitride nanosheets |
| NG | N−doped graphene |
| EBP | exfoliated black phosphorus |
| R-graphyne | rectangular graphyne |
| HWE | hybrid water electrolysis |

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
