# Peer review of "Recent Developments in Two-Dimensional Carbon-Based Nanomaterials for Electrochemical Water Oxidation: A Mini Review"

_catalysts, doi:10.3390/catal14040221_

Round 1

Reviewer 1 Report

Comments and Suggestions for Authors

The current form of the revised manuscript still need more attention. My comments are as follows:

1.      The authors mentioned that the current mini review summarizes the recent research work in regard to the synthetic methods, OER performance, as well as catalytic mechanisms. However, there are no discussions related to the synthesis strategies and catalytic mechanisms. Therefore, please add brief sections related to these important topics.

2.      Recently, several reviews addressing the employment of carbon based electrocatatlysts for OER have been reported such as doi.org/10.1016/j.pmatsci.2020.100717, doi.org/10.1038/s41524-019-0210-3, doi.org/10.1021/accountsmr.1c00190, https://doi.org/10.1002/cctc.201901707, and etc, hence, I invite the authors to clearly highlight the main aim of the current mini review to make it more readable.

3.      The introduction part should be further reinforced. Some new studies related to electrocatalysis-related areas can be cited such as 10.1016/S1872-2067(23)64544-9 and doi.org/10.1016/j.ccr.2023.215405.

4.      In this submission, the authors just summarize some reported studies, however their claim is not clear. The authors should explain the effect of tailoring the compositional and structural features of an electrocatalyst on the OER activity.

5.      Please correct the title No. 5. All abbreviations should be outlined in a table before the introduction part. It is better to add your own Figures (at least one) not only copying pictures from reported works. Authors can construct schematic  illustrations related to OER mechanism and current challenges.

Reviewer 2 Report

Comments and Suggestions for Authors

The manuscript reviews the recent developments of two-dimensional carbon-based nanomaterials and their application for electrochemical water oxidation. The authors discussed the recently developed carbon base catalysts, characterization techniques, and application of precious/nonprecious and non-metal-doped carbon materials for oxygen evolution reactions. The manuscript was prepared in a readable style. Therefore, I recommend the publication of the manuscript on catalysts after some issues being revised:

1.    The first section of this review must be promising, detailing the most recent advances in 2D carbon-based nanomaterial synthesis and functionalization using growth and delamination processes ranging from chemical to electrochemical methods. The design, fabrication, and mechanism investigations over different types should be summarized.

2.    As a review, authors should highlight the structure–activity relationship between specific metals, nonmetals, heteroatoms/defects, and OER catalytic performance within 2D metal-carbon electrocatalysts.

3.    The quality of Figure, 3a, b, and c, is unsatisfactory because the details are simply indistinguishable. Every part of the figures in the article needs to be carefully checked and modified.

4.    As a review, it should discuss the most recent papers that appeared in this area.

5.    Different classes of OER electrocatalysts need to be compared in terms of stability. The stability of 2D carbon nanomaterials electrocatalysts is mentioned as one of the important criteria, more insight into the processes leading to instability and its results will be of interest to readers.

6.    Conclusions need to be improved. Authors should make it brief and consistent with their main discussion. General statements need to be removed. Authors are suggested to be more specific in their conclusions.

7.    Reference/ must match with journal style.

Comments on the Quality of English Language

 Moderate editing of English language required.

Reviewer 3 Report

Comments and Suggestions for Authors

Recent developments of two-dimensional carbon-based nano-materials for electrochemical water oxidation: a mini-review needs to be improved by minor revision. Here are the comments:

1. Please, add the appropriate references at the end of the next sentences in the manuscript:

The excellent performance of Ru-H2O/CC-350 could be ascribed to the formed Ru/RuO2 heterostructure with ample defects of oxygen vacancies.ÌŽ

ÌŽThe 2D morphology of CoRu alloy provides sufficient active sites and the synergistic effect between Co and Ru ensures the good electrocatalytic activities.ÌŽ

ÌŽThe distinguished activity and stability could be attributed to the special hierarchical porosity, high conductivity of N-doped graphene shell, and Ru-enriched surface.ÌŽ

ÌŽThe outstanding electrocatalytic performances are due to the synergistic effect of Co and Ru, the abundant pores of carbon matrix, as well as the junction of CoRu composites with carbon matrix.ÌŽ

ÌŽThe inexpensive GO and Rh nanospheres provide high porosity and active surface area, which ensure the excellent OER performances.ÌŽ

ÌŽThe outstanding OER performances of 2D heterostructure NiS/G-3 could be attributed to the synthetic effect between 2D graphene and non-precious NiS composites.ÌŽ

ÌŽIn general, as a OER electrocatalyst, Ni-Co-P/GDY manifests excellent performance under alkaline conditions, owing to the synergistic effect of Ni-Co-P and GDY.ÌŽ

ÌŽThe good OER performances could be attributed to the synergistic effect between metal dopants and R-graphyne, as well as the increased antibonding characteristics that offer proper adsorption state of O*.ÌŽ

ÌŽThe DFT calculations and experimental results imply that synergistic effect between EBP and NG optimizes the adsorption energies of OER intermediates, which promote the formation of OOH* and finally improve the OER energetics.ÌŽ

ÌŽThe excellent electrocatalytic performances of the carbon-based heterogeneous g-C3N4/rGO could be attributed to the introduction of 2D materials which enhances the electron transfer to the interface between electrode and optimizes the electrocatalytic active sites for OER.ÌŽ

ÌŽThe excellent OER performances could be ascribed to the large amount number of active sites provided by high content of graphitic N. Besides, the employment of 2D graphene affords the support and co-catalyst that improves the OER performances by enhancing the charge transfer.ÌŽ

ÌŽThe excellent OER performances could be ascribed to the sufficient active sites supported by the large surface area and effective charge transfer and separation provided by the 2D S-CN/CN homojunction.ÌŽ

ÌŽThe nanoconfinement offered by graphene multilayers not only ensure the formation of highly active species and thus greatly enhances the electrocatalytic performances for OER.ÌŽ

ÌŽThe outstanding electrocatalytic OER performances could be attributed to the ample active sites and high graphitic degree. Importantly, the confinement environment provided by N-doped carbon nonosheets makes the active sites easier to approach the electrolyte.ÌŽ

ÌŽThe good electrocatalytic OER performances could be ascribed to the NaCl confinement that availably hinders the intermediates and produces the holed NCNs.ÌŽ

2. What does mean ZIF-derived? Please, add an explanation for the noted abbreviation

3. Why do the authors present only ruthenium catalysts in Table 1?

4. Please, add the appropriate references in the first paragraph of the 5. Summary and Outlook section.

Reviewer 4 Report

Comments and Suggestions for Authors

In this review, the authors review the recent development of 2D carbon-based nanomaterials for alkaline water splitting. The doping strategies, current challenges, opportunities, and perspective are discussed. However, some important issues must be solved before acceptance.

1.     To emphasize the importance of green hydrogen production and improve the readability of the manuscript, the authors should discuss about the global hydrogen requirement and economics. Besides, this review focuses on the alkaline water splitting, the discussion about the related applications should be included in the introduction part. The authors can refer to this work 10.1039/D3EE02695G for these points.

2.     The surface area and stability of the listed examples should also be included and compared. Also, the types of the 2D carbon-materials involved in this review should be mentioned prior to the detailed content. Furthermore, the comparisons between 2D carbon-based materials and other 2D materials should be involved, e.g., layered 2D oxide like 10.1016/j.jechem.2023.03.033. This will help enhance the importance of 2D carbon-based materials in green hydrogen production.

3.     The structural schematic diagrams of each type 2D carbon-based materials should be presented to help understand the structure.

4.     In the perspective, the development of applications and the activity/stability descriptors should also be included.

Comments on the Quality of English Language

Minor editing of English language required

Round 2

Reviewer 1 Report

Comments and Suggestions for Authors

The authors have adjusted all the comments. I suggest the publication of this paper

Reviewer 2 Report

Comments and Suggestions for Authors

The author has successfully addressed all the comments. Moreover, author has made significant changes in the manuscript to fulfill the reviewer's requirement. The revised manuscript is now well-readable and eligible to be published in this journal.